

# Trends in Water Vapor in North America Based on GNSS observation and ERA5 reanalysis

Yuling Zhao[1], Ce Zhang[1], Shuaimin Wang[1], Yujing Xu [1], Hong Yu[1]

[1]School of Mining and Geomatics Engineering, Hebei University of Engineering, Handan, Hebei, 056038, China

Correspondence: Shuaimin Wang (wsm1949@sina.com)

Correspondence: Yujing Xu (xuyujing410@163.com)

**Abstract.** Atmospheric precipitable water vapor (PWV) is a very important meteorological factor for weather forecasting and climate change monitoring. GNSS observation and ERA5 reanalysis for the period of 2010-2022 are used to analyse the overall and seasonal distribution and interdecadal trend of PWV in North America. The results indicate that GNSS and
ERA5 are in good agreement between the distribution and interdecadal trend of PWV. The overall PWV change from 2010 to 2022 shows an obvious upward trend. The mean PWV is less than 20 mm in most regions of North America except for the Southeast where mean PWV is more than 30 mm. The change trend and correlation of PWV and temperature in North America from 1940 to 2022 based on ERA5 reanalysis data are analysed. The results show that the PWV increases more significantly with the interdecadal trend of 0.17 mm decade$^{-1}$. The results illustrate that there is a significant correlation
between PWV and temperature with the correlation coefficient of 0.96, and that there are some differences between the actual increase and ideal increase of water vapor content derived from Clausius-Clapeyron equation for every 1 K increase in North America. In addition, the results also show a significant increase of PWV during strong Ei Nino events and a significant decrease of PWV during strong La Nino events, thereby indicating that Ei Nino and La Nino events have an important influence on the change of PWV in North America.

**1 Introduction**

Water vapor, the most active component of the atmosphere, is critical to many atmospheric processes, including the form of clouds, rain, and snow as well as the global energy conversion and climate change, even though it accounts for less than 5% of the atmosphere(Mo et al., 2021; Zhao et al., 2019a; Huang et al., 2021). Common precipitable water vapor(PWV) products include radiosondes PWV, remotely sensed PWV, Global Navigation Satellite System (GNSS) PWV, and
reanalysis PWV data(Zhang et al., 2019). Various PWV products have their own characteristics. Radiosonde observation has the advantages of a long observation history and stable accuracy, and is often used as reference data for other water vapor products. The disadvantages of the radiosonde stations are sparse spatial distribution which can't fully characterize the spatial distribution of water vapor in the study area, and the low temporal resolution of 2 to 4 times a day(Chang et al., 2015;



Zhao et al., 2019b). Near infrared remote sensing provides atmospheric water vapor information with high spatial resolution,

but the PWV is strongly influenced by the observation conditions, with obvious systematic biases in the products and low data availability(Liang et al., 2014). GNSS PWV is obtained by solving the ground-based GNSS observations, which has the characteristics of high precision, high temporal resolution, and all-weather (Minghua, 2023; Vaquero-Martínez et al., 2018). The reanalysis PWV data have a uniform spatial distribution and temporal resolution, but its applicability needs to be examined using high-precision ground-based observations(Wang et al., 2017).

Many scholars have conducted relevant research on the global PWV trend. Chen and Liu (2016) analysed the global PWV trend from 1979 to 2014 using the ERA-Interim dataset. The results indicate that the PWV trend is increasing with an interval of $0.61 \pm 0.33\%$ decade$^{-1}$. Ren et al. (2023) compared the global PWV trends based on the ERA5 and JRA55 datasets from 1958 to 2020. The results demonstrate that the PWV values are increasing by an extent of $0.19 \pm 0.01$ mm $(1.15 \pm 0.31\%)$ decade$^{-1}$ and $0.23 \pm 0.01$ mm $(1.45 \pm 0.32\%)$ decade$^{-1}$ for ERA5 and JRA55, respectively. Wang et al. (2020)

evaluated the accuracy of PWV values of five reanalysis products using PWV values from 268 GNSS stations around the world from 2016 to 2018, and the results illustrate that the ERA5 reanalysis PWV product performs the best among the five reanalysis products. Water vapor trends in some regions and countries have been well analysed. Sarkar et al. (2023) analysed the PWV trend in India from 1980 to 2020 based on ERA5 reanalysis. The results reveal that the increasing interdecadal trend of PWV reaches 0.09 mm year$^{-1}$. Nilsson and Elgered (2008) analysed the atmospheric water vapor trends over Finland

and Sweden using 10 years of observations from 33 GNSS stations and obtained a linear trend in the range of -0.2 ~ 1.0 kg m$^{-2}$ decade$^{-1}$. Bernet et al. (2020) investigated the water vapor trends over Switzerland using water vapor radiometer, FTIR (Fourier Transform Infrared) and GNSS-PWV information and recovered a typical increase in water vapor in the range of 2% to 5% decade$^{-1}$. Alshawaf et al. (2018) estimated 113 time series of GNSS-PWV in the German region for the period 2010-2019 using the Theil-Sen method, and the results reveal that the PWV in the German region shows a trend of -0.15 to 0.23

mm decade$^{-1}$. Peng et al. (2017) used ERA-Interim data and found that the upward trend of PWV in northern China is 0.1~1.2 mm decade$^{-1}$. Xu et al. (2020) used ERA-Interim and JRA55 data to acquire an increasing trend of $0.12 \times 10^6$ kg s$^{-1}$ year$^{-1}$ on the Tibetan Plateau from 1979 to 2018.

A number of studies of other climatic phenomena have been carried out in the North American region. Van Wijngaarden and Isaac (2012) studied water vapor pressure and temperature at the most of the 309 monitoring stations in

North America and found that water vapor pressure increased with a trend of +0.07 h Pa or +0.7% decade$^{-1}$, and temperature increased with a trend of 0.2°C decade$^{-1}$. Li et al. (2020) analysed the spatial and temporal distributions of extreme values of environmental parameters and weather-scale features of severe local storms (SLS) in North America from 1980 to 2014 in ERA5 reanalysis data and CAM6 historical simulation data. The results demonstrate that both ERA5 and CAM6 are capable of reproducing the SLS environment as well as generating its weather-scale features and transient events. Taszarek et al.

(2021) compared the observations from 232 radiosonde stations in Europe and North America with the proximal vertical profiles of ERA5 and MERRA-2. The results show that ERA5 has higher correlation and lower mean absolute error than MERRA-2. Urraca and Gobron (2023) assessed the stability of the ERA5 and ERA5-Land reanalysis products from 1950 to



2020 and the National Oceanic and Atmospheric Administration Climate Data Record (NOAA CDR) from 1966 to 2020 using 527 ground stations from 1950 to 2020 as reference data. The study discovers that a negative trend in snow depth of -

0.9 ~ -1.6 cm decade$^{-1}$ has been observed in the eastern United States and most of Canada. Torralba et al. (2017) used ERA-Interim reanalysis data for the period 1980-2015 to study wind speed trends, and found a clear upward trend in the spring in the northwestern region of North America.

In summary, on the one hand, a large number of studies have been conducted on the changes trend of water vapor in the worldwide or other countries and regions, and water vapor trends are different in different regions. On the other hand,

although there have been many studies on the trend of other meteorological factors in North America, there are few studies on the long-term trend of PWV in the North American region. In addition, ERA5 reanalysis PWV performs the best accuracy among the global reanalysis products with a high spatial resolution and a long history data, and GNSS-PWV has the advantages of high precision, high temporal resolution, and all-weather. Therefore, the distribution and trend of water vapor from 2010 to 2022 and from 1940 to 2022 over North America are analysed and discussed based on GNSS-PWV and

ERA5-PWV in this study.

The paper is structured as follows. Section 2 describes the dataset and methodology in detail. Section 3 firstly compares and analyses the distribution, and interdecadal trends of water vapor from 2010 to 2022 based on GNSS and ERA5 reanalysis in the North American region. Secondly, the distribution of water vapor, interdecadal variability in different seasons and interdecadal trends in water vapor from 1940 to 2022 are investigated. The correlation between PWV and temperature, and

the effect of ENSO climate factors on water vapor variability are then analysed. Section 4 concludes and summarizes the paper.

## 2 Data and Methodology

### 2.1 GNSS PWV

SuomiNet is primarily a GNSS network in North America, which provides high accuracy and time resolution GNSS-PWV

data which with the root mean square error (RMSE) of less than 2 mm and temporal resolution of 30 minutes, which are estimated from GNSS zenith tropospheric delays and conversion factors(Ware et al., 2000). The SuomiNet network consists of about 800 GNSS stations. The missing data and outliers in the GNSS-PWV time series are unavoidable due to GNSS receiver failures or poor observing conditions, therefore, in this paper, we use three times standard deviation method to remove the outliers, and the missing data are processed by the linear interpolation method. Finally, we obtain 110 GNSS

stations with the continuous PWV data from 2010 to 2022, which are shown in Fig. 1.



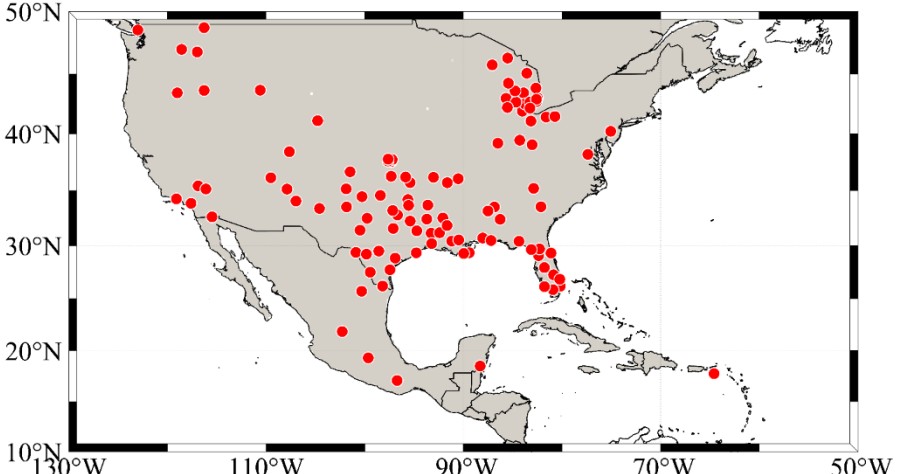

**Figure 1. Location distribution of GNSS station.**

## 2.2 ERA5 reanalysis PWV products

Atmospheric reanalysis products have been widely used in various meteorological studies(Parracho et al., 2018). The
European Centre for Medium-Range Weather Forecasts (ECMWF) has released ERA5 reanalysis, the latest fifth-generation
reanalysis of the global weather and climate record for all types of atmospheric, land and ocean surface meteorological
parameters since 1940 (Hersbach et al., 2020). The ERA5 reanalysis products have a spatial resolution of $0.25° \times 0.25°$ and a
temporal resolution of 1 hour. In this paper, PWV and temperature from ERA5 reanalysis dataset covering the North
American region from 1940 to 2022 are used.

## 2.3 ENSO indexes

The ENSO indexes (El Nino 3.4 area, 5°N-5°S, 120°W-170°W) use the Southern Oscillation Index (SOI) and the Oceanic
Nino Index (ONI), which can be obtained from the National Oceanic and Atmospheric Administration (NOAA) of the
United States. Through the ocean region, ENSO is the most commonly monitored by the ONI, which is determined from
mean Sea Surface Temperature (SST) anomalies in a box-shaped region in the east-central tropical Pacific known as El Nino
3.4(Xu et al., 2023). In the atmosphere, ENSO is monitored by the SOI, a measurement of atmospheric circulation based on
the difference in atmospheric pressure levels measured between Darwin and Tahiti (Trenberth and Caron, 2000).

## 2.4 Theil-Sen trend estimation

Theil-Sen trend estimation (Sen, 1968; Theil, 1950) is a nonparametric statistical method for calculating linear trends for the
median slope of all pairwise point combinations, and is insensitive to outliers. Therefore, the Thile-Sen trend estimation is
used to calculate the water vapor trend in this paper. The Theil-Sen formula is as follow:





$$Q_i = \frac{x_j - x_k}{j - k} \quad i=1, \cdots N \tag{1}$$

where $x_j$ and $x_k$ are the water vapor values ($j > k$) and $N = \frac{n(n-1)}{2}$ at time $j$ and time $k$ respectively. Arranging the $N$

values of $Q_i$ from smallest to largest, the median Sen's slope is estimated as:

$$Q_{med} = \begin{cases} Q_{[(N+1)/2]} & N \text{ for the odd number} \\ \dfrac{Q_{[N/2]} + Q_{[(N+2)/2]}}{2} & N \text{ for the even number} \end{cases} \tag{2}$$

where $Q_{med}$ responds to the steepness of the trend of the data. $Q_{med}$ greater than zero indicates that water vapor is an

upward trend, and vice versa for a downward trend.

**2.5 Correlation test**

Pearson Correlation Coefficient (PCC) (Lee Rodgers and Nicewander, 1988) is used to represent the linear correlation

between $PWV$ and $Temperature$, and the correlation results are tested for significance using t-test (Student's test),

with the significance level taken as α=0.05.

$$R = \frac{\sum_{i=1}^{n}(PWV_i - \overline{PWV})(T_i - \overline{T})}{\sqrt{\sum_{i=1}^{n}(PWV_i - \overline{PWV})^2}\sqrt{\sum_{i=1}^{n}(T_i - \overline{T})^2}} \tag{3}$$

where $PWV_i$ and $T_i$ are the $PWV$ values and the $Temperature$ values at time $i$, and $\overline{PWV}$ and $\overline{T}$ are the mean

values of $PWV$ and $Temperature$, respectively.

**3 Results and Discussion**

**3.1 Mean distribution and trends of PWV from 2010 to 2022 based on GNSS and ERA5 in North America**

The PWV data based on GNSS observation and ERA5 reanalysis from 2010 to 2022 in North America is used to analyse the

mean distribution and trend of PWV.

**3.1.1 Mean distribution of PWV from 2010 to 2022 based on GNSS and ERA5 in North America**

The comparison of mean PWV distributions in different periods in North America from 2010 to 2022 is illustrated in Fig. 2

and Fig. 3 based on GNSS observation and ERA5 reanalysis.



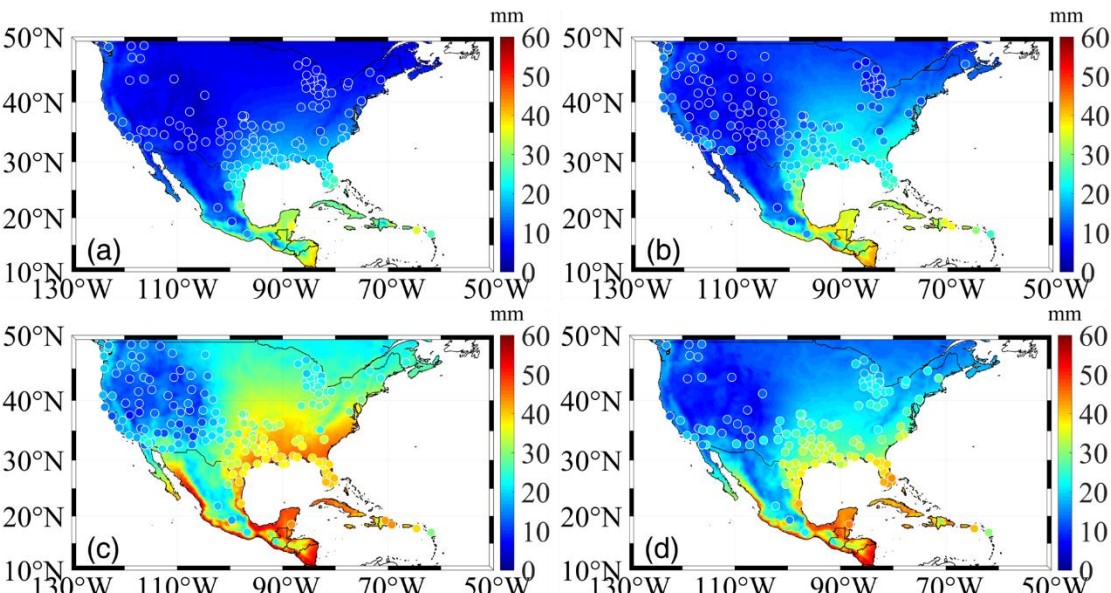

**Figure 2. Mean distribution of PWV from 2010 to 2022 based on GNSS and ERA5 in North America in (a) winter, (b) spring, (c) summer and (d) autumn.**

Fig. 2 indicates that there is a good agreement between GNSS-PWV and ERA5-PWV in different seasons. Fig. 2a
shows that the mean PWV values in winter are mostly below 20 mm except near the Gulf of Mexico where PWV values are
above 30 mm and in the western region of the Caribbean Sea where PWV can reach 40 mm. However, Fig. 2c reveals that
the mean PWV values in summer are mostly above 20 mm except for those near the Rocky Mountains which are below 20
mm, especially in the Gulf of Mexico and the Caribbean Sea. In comparison, mean PWV in spring and autumn is
intermediate, ranging from about 30 mm near the Gulf of Mexico to less than 20 mm near the Cordillera system. The
western region of the study area is influenced by the Cordillera system and the mean PWV is below 20 mm in each season.
Near the Gulf of Mexico, the mean PWV is consistently higher than 20 mm in each season in particular around the
Caribbean Sea are from 30 mm to 50 mm, but the PWV decreases at the same latitude due to the influence of the Mexican
Plateau and the Sierra Madre. In other regions, including the central and the east, the mean PWV has an obvious seasonality.



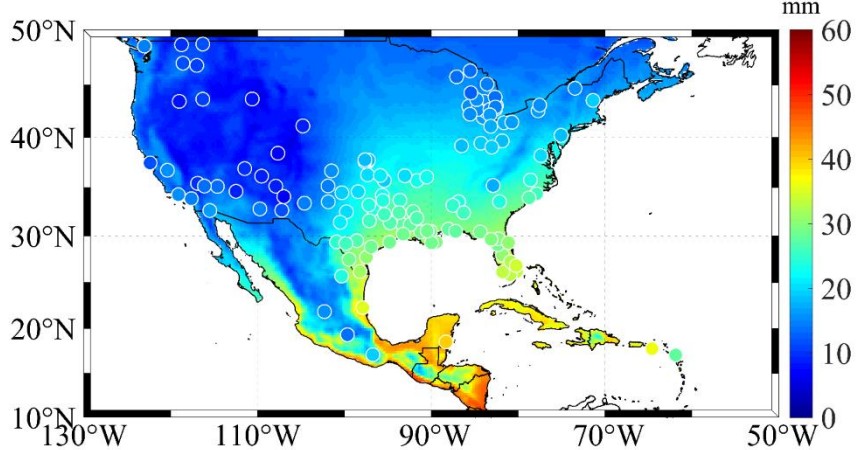

**Figure 3. Mean distribution of PWV from 2010 to 2022 based on GNSS and ERA5 in North America.**

Fig. 3 indicates that there is a good agreement between the annual mean GNSS-PWV and ERA5-PWV distribution. The mean PWV in most regions of North America is less than 20 mm from 2010 to 2022. However, the mean PWV values are around 30 mm near the Gulf of Mexico and especially above 40 mm near the Caribbean Sea. The mean PWV at the same latitude is decreased due to the influence of the Mexican Plateau and the Sierra Madre. The mean PWV near the Cordillera system in the west is below 10 mm, and the mean PWV in the centre and east decreases with the increasing latitude.

**3.1.2 Trends of PWV from 2010 to 2022 based on GNSS and ERA5 in North America**

The distributions of seasonal and overall trends of PWV in North America from 2010 to 2022 based on GNSS observation and ERA5 reanalysis are illustrated in Fig. 4 and Fig. 5.





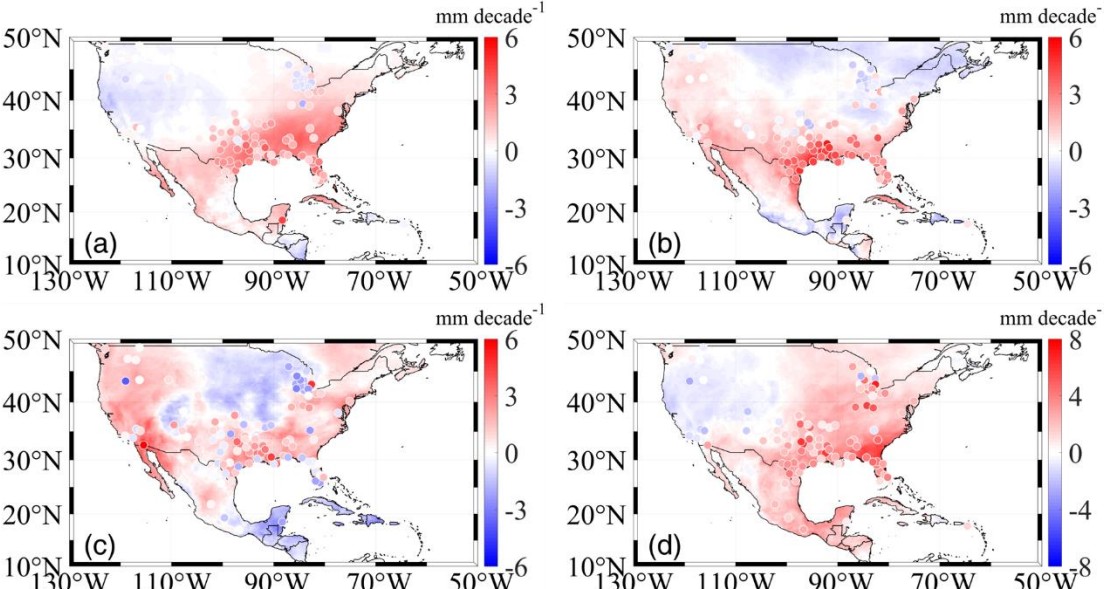

**Figure 4. Trend distribution of PWV from 2010 to 2022 based on GNSS and ERA5 in North America in (a) winter, (b) spring, (c) summer and (d) autumn.**

Fig. 4 shows the trend of PWV in the western region of North America does not significantly decrease in autumn and winter, but increases in spring and summer. The trend of PWV in the southeastern region of America tends to rise in every season. PWV in the northeast of North America shows a downward trend in spring, while it shows an insignificant upward

trend in other seasons.

Fig. 4a indicates that there are some significant increases in PWV along the Atlantic coast and the Gulf of Mexico with the PWV trend of above 3 mm decade$^{-1}$ in winter, and there is a decrease in PWV near the Rocky Mountains with the PWV trend of around -1.5 mm decade$^{-1}$. Fig. 4b demonstrates that there is a significantly increasing PWV trend of around 3 mm decade$^{-1}$ in spring from 25°N to 30°N. Fig. 4c displays that the interdecadal trend of PWV in summer increases significantly

in the most areas, especially in the Gulf of California, where the PWV increase is more significant with the trend above 4 mm decade$^{-1}$, and decreases significantly in the Great Plains, with the trend around -2 mm decade$^{-1}$. Fig. 4d shows that the interdecadal trend of PWV is significantly increasing in most areas in autumn, especially around the Gulf of Mexico. Fig. 4d shows that the interdecadal trend of PWV is significantly increasing in most areas in autumn, especially around the Gulf of Mexico, with the trend above 4 mm decade$^{-1}$. However, PWV near the Rocky Mountains declines with the trend around -1

mm decade$^{-1}$. The interdecadal trends in PWV in the Rockies show an insignificant downward trend in autumn and winter, but an upward trend in spring and summer. In the Gulf of Mexico, the interdecadal trends in PWV are increasing in every season. The interdecadal PWV trends from 30°N to 50°N are significantly affected by the seasonality, but those from 10°N to 30°N are not significantly affected by the seasonality.



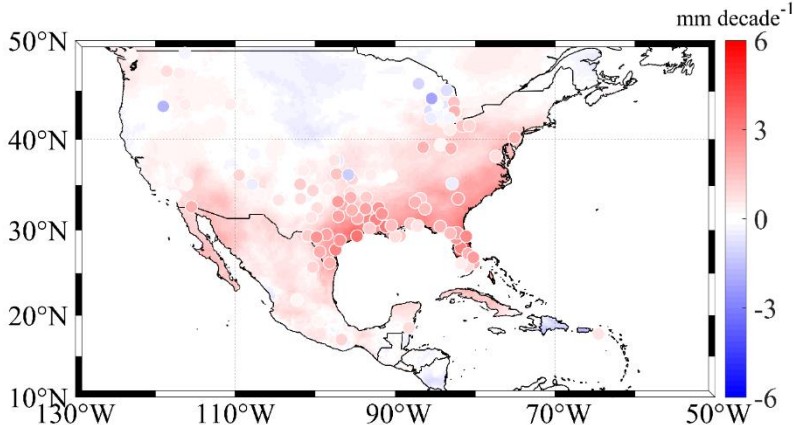

**Fig. 5. Trend distribution of PWV from 2010 to 2022 based on GNSS and ERA5 in North America.**

Fig. 5 illustrates that the most trends of ERA5-PWV and GNSS-PWV in North America from 2010 to 2022 exhibit good agreement. However, there are some differences between ERA5-PWV and GNSS-PWV in the Atlantic coastal region, probably since there are fewer ground-based observations and remotely sensed PWV observations of poor quality used for assimilation to ERA5 reanalysis in coastal region, which in turn affects the ERA5-PWV accuracy in the coastal region. At high altitudes and regions with complex topography, there are discrepancies between ERA5-PWV and GNSS-PWV due to factors such as horizontal gradient and few number of ground observation stations, which may compromise the accuracy of ERA5-PWV. In addition, although the vacant GNSS data is linearly interpolated, GNSS data discontinuity and inhomogeneities in the GNSS equipment maybe still have some influences on the PWV trend difference between GNSS and ERA5.

PWV change near the Gulf of California and southeast North America indicates an upward trend, and PWV in central and northern North America reveals an insignificant downward trend from 2010 to 2022. However, PWV change in other regions of North America is not obvious.

**3.2 PWV trend from 1940 to 2022 based on ERA5 in North America**

The PWV trends above are estimated over a fairly short 13-year period (2010-2022), and maybe will be affected by the large single events such as ENSO. Therefore, in this section we analyse the trends in PWV over the North America region from 1940 to 2022. Fig. 6 shows the interdecadal trend of PWV for each season.





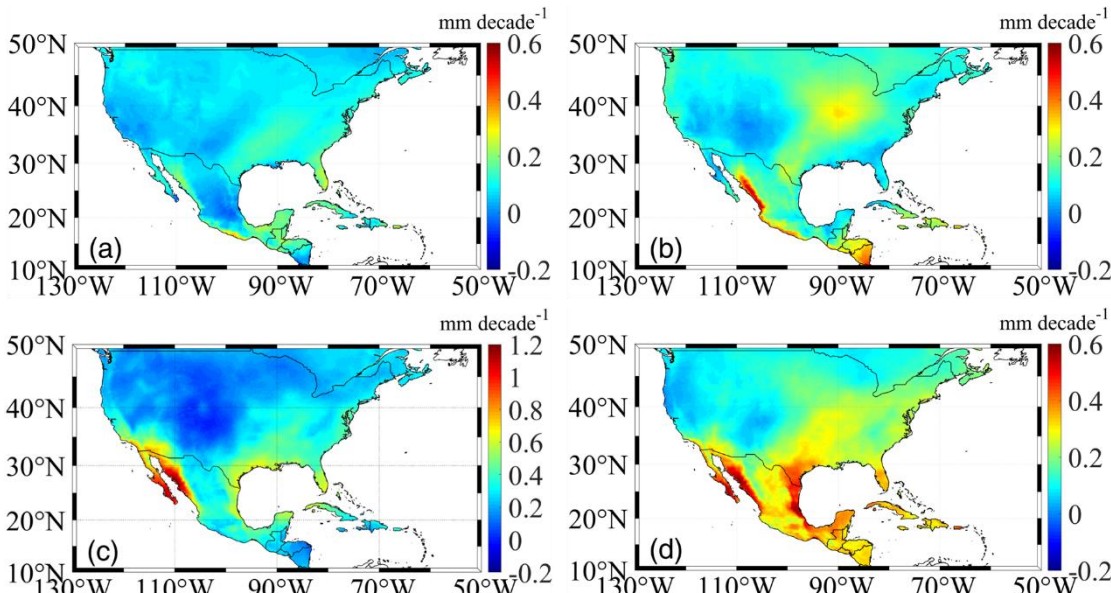

**Figure 6. Trend distribution of PWV from 1940 to 2022 based on ERA5 in North America in (a) winter, (b) spring, (c) summer and (d) autumn.**

Fig. 6 indicates that the interdecadal trend of PWV for all seasons is mostly upward. PWV trends in North America have obvious seasonality. Fig. 6a shows that all the interdecadal trends of PWV in winter are below 0.2 mm decade$^{-1}$. Fig. 6b reveals that, the PWV trends are below 0.3 mm decade$^{-1}$ in most regions of North America. However, the interdecadal trend of PWV in spring increases significantly in the Gulf of California and the Great Plains region, where the PWV trends are above 0.4 mm decade$^{-1}$ and around 0.3 mm decade$^{-1}$, respectively. Fig. 6c demonstrates that the interdecadal trend of PWV

in summer increases significantly in the Gulf of California and the Gulf of Mexico, where PWV trends are above 0.8 mm decade$^{-1}$ and around 0.5 mm decade$^{-1}$, respectively. Fig. 6d exhibits that the interdecadal trend of PWV in autumn increases with the trend above 0.2 mm decade$^{-1}$ in the Gulf of California and the Gulf of Mexico, while the PWV trend is below 0.2 mm decade$^{-1}$ near the Rocky Mountains.



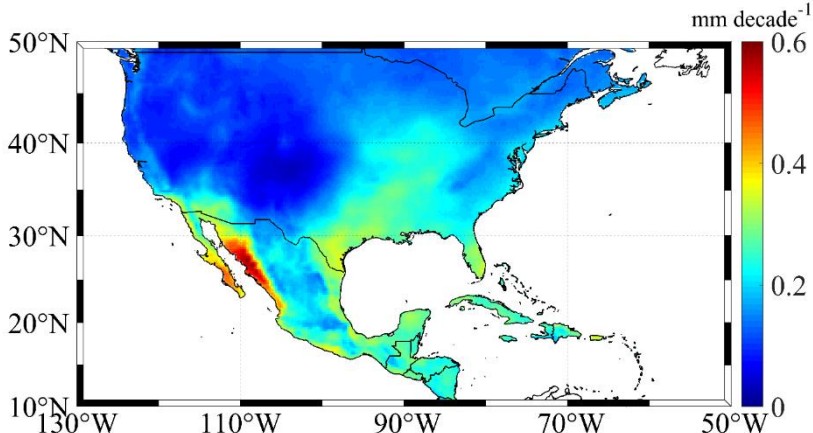

**Figure 7. Trend distribution of PWV from 1940 to 2022 based on ERA5 in North America.**

The Theil-Sen trend estimates from 1940 to 2022 in North America are shown in Fig. 7. Fig. 7 indicates that the whole interdecadal trend of PWV is rising in North America from 1940 to 2022. There is a significant upward trend of PWV, especially in the most of the southern regions of North America, with the fastest rising inter-decadal trend of around 0.60 mm decade$^{-1}$ near the Gulf of California. However, there is an insignificant interdecadal trend of around 0 mm decade$^{-1}$ near

the Rocky Mountains.

### 3.3 Analysis of the relationship and trend of PWV and temperature from 1940 to 2022 based on ERA5 in North America

The relationship and trend between PWV and temperature in the North American region from 1940 to 2022, from 1940 to 1980, and from 1980 to 2022 are shown in Fig. 8.




**Figure 8. Relationship between PWV and temperature based on ERA5 in North America (a) from 1940 to 2022, (b) from 1940 to 1980, (c) from 1981 to 2022**



Fig. 8a indicates that the fluctuation of PWV and temperature is insignificant during 1940-1980, but PWV and temperature fluctuate markedly after 1980. Therefore, the change of PWV and temperature is divided into two segments before and after 1980. The results reveal that the correlation coefficients between PWV and temperature in North America are 0.96 for both 1940-1980 and 1980-2022, indicating that PWV and temperature are well correlated.

On the one hand, Fig. 8a and 8c illustrates that an increase in water vapor, as a greenhouse gas, leads to a rise in temperature. On the other hand, ideally the water vapor content should increase by 6.8% for every 1K increase in temperature according to the Clausius-Clapeyron equation. In fact, Fig. 8a and 8c show that the increase of 8.24% in water vapor for every 1K increase in temperature from 1940 to 2022 is greater than the ideal increase of water vapor from Clausius-Clapeyron, and that the increase of 4.47% in water vapor for every 1K increase in temperature from 1980 to 2022 is less than those from Clausius-Clapeyron. The results illustrate that there are some differences between the actual increase and ideal increase of water vapor content for every 1 K increase in North America. Fig. 8b indicates that the increase of water vapor content even tends to 0 from 1940 to 1980 when the temperature increases insignificantly, which is very inconsistent with the law of Clausius-Clapeyron equation.





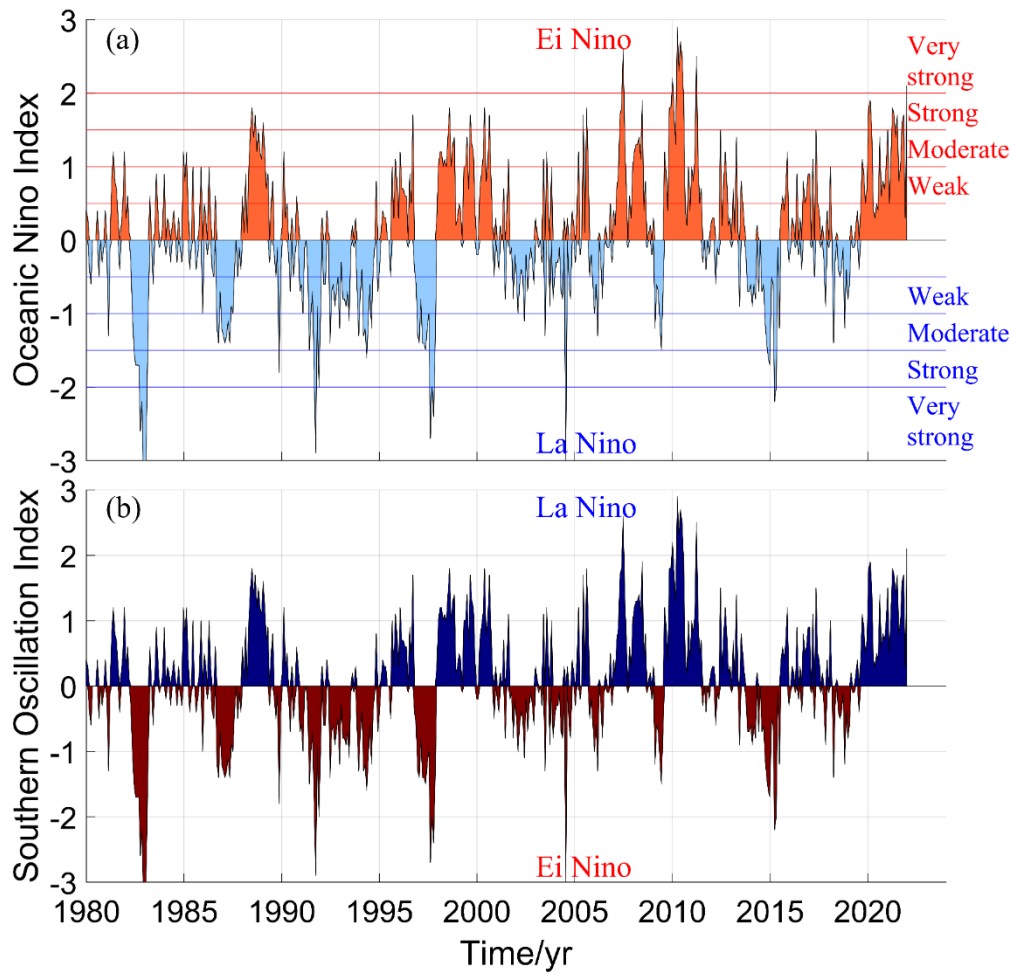

**Figure 9. ONI and SOI from 1980 to 2022**

Fig. 8c and Fig. 9 demonstrate that super or strong class El Niño events occur during 1982-1983, 1987-1988, 1991-1992, 1997-1998, and 2015-2016, when there are some significant increases in PWV. La Niña events of strong magnitude

occur during 1988-1989, 1998-2000, 2007-2008, and 2010-2011, when the PWV values decrease significantly, thereby indicating that Ei Nino and La Nino events have an important influence on PWV in North America.

## 4 Conclusion

Water vapor is a crucial component of the Earth's climate system, playing a vital role in atmospheric changes. The mean PWV distributions and the interdecadal trends of PWV are analysed in the North American region using GNSS-PWV and

ERA5-PWV data for the period of 2010-2022 and 1940-2022.



Firstly, the mean PWV distribution and the interdecadal trend of PWV are analysed in the North American region using GNSS-PWV and ERA5-PWV data for the period of 2010-2022. The GNSS-PWV and ERA5-PWV in North America are in good agreement on both mean distribution and interdecadal trend of PWV. The mean PWV is less than 20 mm in the most regions of North America except near the Gulf of Mexico and near the Caribbean Sea where mean PWV is more than 245 30 mm. The interdecadal trends in PWV in the Rockies show an insignificant downward trend in autumn and winter, but an upward trend in spring and summer. In the Gulf of Mexico, the interdecadal trend in PWV is increasing in every season. Influenced by the Mexican Plateau and the Sierra Madre, the mean PWV decreases at the same latitude. The interdecadal trend of PWV from 30°N to 50°N has obvious seasonality, while the interdecadal trend of PWV from 10°N to 30°N is insignificantly affected by seasonality.

Secondly, the ERA5-PWV data from 1940 to 2022 are used to analyse the PWV trends in different seasons as well as the interdecadal trends in North America. Most of interdecadal trends of PWV in every season are below 0.2 mm decade$^{-1}$ except in summer when most of interdecadal trends of PWV are below 0.4 mm decade$^{-1}$. The inter-decadal trend near the Gulf of California is very significant with the trend around 0.60 mm decade$^{-1}$, while there is an insignificant interdecadal trend near the Rocky Mountains with the trend around 0 mm decade$^{-1}$. The overall increasing trend in water vapor during 255 1980-2022 is more significant at 0.13 mm decade$^{-1}$. The overall increasing trend in water vapor during 1940-1980 is insignificant and closes to 0 mm decade$^{-1}$.

Finally, we analyse the correlation between PWV and temperature from 1940 to 2022 based on ERA5 data in North America, and explore the influence of ENSO on PWV change. The correlation coefficients between PWV and temperature within the North American region from 1940 to 2022, from 1940 to 1980, and from 1980 to 2022 all are 0.96, indicating a 260 good correlation between PWV and temperature. The results indicate that there are some differences between the actual increase and ideal increase of water vapor content derived from Clausius-Clapeyron equation for every 1 K increase in North America. The increase of water vapor content even tends to 0 from 1940 to 1980 when the temperature increases insignificantly, which is very inconsistent with the law of Clausius-Clapeyron equation. The finds that PWV increases significantly during strong Ei Nino events and decreases significantly during strong La Nino events, which indicates that 265 PWV change is obviously affected by Ei Nino and La Nino events.

**Data availability**. The European Centre for Medium-Range Weather Forecasts (ECMWF) provide the ERA5 reanalysis at https://cds.climate.copernicus.eu. The University Corporation for Atmospheric Research (UCAR) provide the GNSS observation at https://www.cosmic.ucar.edu. The National Oceanic and Atmospheric Administration (NOAA) provide SOI and ONI at https://origin.cpc.ncep.noaa.gov.

**Author contribution.** Yuling Zhao: Project administration, Funding acquisition, Writing – review & editing. Ce Zhang: Methodology, Investigation, Software, Writing original draft, Writing – review & editing. Shuaimin Wang: Conceptualization, Methodology, Resources, Data curation, Supervision, Project administration, Funding acquisition, Writing – review & editing. Yujing Xu and Hong Yu: Writing – review & editing

**Competing interests.** The contact author has declared that none of the authors has any competing interests.



**Disclaimer.** Publisher's note: Copernicus Publications remains neutral with regard to jurisdictional claims made in the text, published maps, institutional affiliations, or any other geographical representation in this paper. While Copernicus Publications makes every effort to include appropriate place names, the final responsibility lies with the authors.

**Financial support.** This work was supported by the Natural Science United Foundation of Hebei Province in China (E2020402086), Natural Science Foundation of Hebei Province in China (D2023402024).

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
