# Peer review of "Trends in Water Vapor in North America Based on GNSS observation and ERA5 reanalysis"

_EGUsphere, 2023_

## Author Comment (AC1)

**Reviewer #1: https://doi.org/**10.5194/egusphere-2023-2508-RC1**

Atmospheric water vapor is an important greenhouse in the atmosphere, and has strong feedback to the global warming, making it a critical element for climate analysis. This work used both GNSS and ERA5 PWV to investigate the distribution and long-term changes of PWV over the north America, and tried to investigate the relationship with temperature changes and ENSO index. Some conclusions were drawn from the work, but in my opinion, the investigations and discussions lack profundity where most of discussions are only simple descriptions without necessary explanations, making it insufficient for publication on ACP.

**Reply:** Thanks very much for your comments and helpful suggestions. It is highly insightful and enabled us to greatly improve the quality of our manuscript. All the suggestions are accepted and the revisions are highlighted in the revised manuscript. Some important discussions and detailed explanations have been added in the revised manuscript. The following problems are explained.

The Theil-Sen trend estimates from 1940 to 2022 in North America are shown in Fig. 8. Fig. 8 (a) indicates that the whole interdecadal trend of PWV is rising in North America from 1940 to 2022. Meanwhile, in Fig. 8 (b), you can see that the trend of PWV remains unchanged east of the Rocky Mountains. The other regions are all experiencing significant increases, and most of them are experiencing extremely significant increases, with the fastest rising inter-decadal trend of around 0.60 mm decade-1 near the Gulf of California. The main reasons of PWV trend increase include two aspects. On the one hand, as global atmospheric and sea surface temperatures rise, sea surface evaporation increases. On the other hand, the global wind speed is increasing, and large amounts of water vapor are transported to the North American continent. The Rocky Mountains on the west side and Appalachian Mountains of the North American continent block water vapor from the Pacific Ocean, so the increase in water vapor is concentrated in the southwest of the Rocky Mountains and in the southeast of Appalachian Mountains.

Fig. 11 (a) demonstrates that super or strong class El Niño events occur during 1982-1983, 1987-1988, 1991-1992, 1997-1998, and 2015-2016 and that PWV residual increases significantly to above 1.5mm from six to ten months after El Nino events. However, PWV do not rise significantly after the extremely strong ENSO event of 1991-1992, probably due to the 1991 volcanic eruption in the Philippines resulting in the decrease of global temperature. The PWV values decrease significantly is reduced by 1.5~3mm after strong La Niña events occur during 1988-1989, 1998-2000, 2007-2008, and 2010-2011. The results indicate that ENSO events have an important influence on PWV change. There is no significant change in PWV after individual ENSO event and La Niña events, which may be due to the superimposed effects of the Atlantic Oscillation and the Arctic Oscillation. Figs. 11 (b) and (c) show that ENSO-induced water vapor changes range from 3 to 8 mm and from 1 to 2 mm in the tropics (10° N ~ 23.5° N) and temperate zones (23.5° N ~ 50° N), respectively. The results

show that the magnitude of ENSO influence on PWV in tropics is higher than that in temperate zone. Figs. 11 (b) and (c) show the response time of PWV to ENSO is shorter in the tropics than temperate zones. The findings indicate that the effect of ENSO on PWV is greater in the tropics than in temperate zones.

Another main flaw is the lack of uncertainty for trend estimates.
**Reply:** Thank for your suggestions. The uncertainty for trend estimates has been added in the revised manuscript.
The Mann-Kendall method is less affected by outliers and missing values because its statistics are based on the sign of the difference, rather than directly on the value of the random variable. Therefore, the Mann-Kendall method is used to analyze the uncertainty and significance for PWV trend estimates [1].
[1] Mann H B . Non-Parametric test against trend[J]. Econometrica, 1945, 13(3): 245-259.
The basic idea is to determine the change trend of time series by calculating the Z-value of the standard normal distribution statistic. The calculation method is as follows:

$$S = \sum_{i=1}^{n-1} \sum_{j=i+1}^{n} sgn(x_j - x_i) \tag{3}$$

Where $n$ represents the sample number of PWV sequence. $sgn(x_j - x_i)$ can be calculated by the following formula:

$$sgn(x_j - x_i) = \begin{cases} +1, & x_j > x_i \\ 0, & x_j = x_i \\ -1, & x_j < x_i \end{cases} \tag{4}$$

When $n \geq 8$, $S$ close to a normal distribution. The sequence variance is calculated as follows:

$$V(S) = \frac{n(n-1)(2n+5)}{18} \tag{5}$$

Then the statistic Z can be obtained:

$$Z = \begin{cases} \dfrac{S-1}{\sqrt{V(S)}}, & S > 0 \\ 0, & S = 0 \\ \dfrac{S+1}{\sqrt{V(S)}}, & S < 0 \end{cases} \tag{6}$$

Z is a normalized statistic of the Mann-Kendall trend test used to calculate statistically significant trends, with positive or negative values of Z representing an upward or downward trend, respectively [2]. There are seven PWV trend grades for Z: Z $\geq$ 2.56 indicating an extremely significant increase trend of PWV, 1.96$\leq$Z<2.56 indicating a significant increase trend of PWV, 1.64$\leq$Z<1.96 indicating an non-significant increase trend of PWV, -1.64<Z<1.64 indicating an unchanged trend of PWV, -1.96<Z$\leq$-1.64 indicating an non-significant decrease trend of PWV, -2.56<Z$\leq$-1.96 indicating a

significant decrease trend of PWV, and Z≤-2.56 indicating an extremely significant decrease trend of PWV. Meanwhile, ±2.56 corresponds to a 99% confidence level, ±1.96 corresponds to a 95% confidence level, and ±1.64 corresponds to a 90% confidence level.

[2] Barry A A, Caesar J, Klein Tank A M G, et al. West Africa climate extremes and climate change indices[J]. International Journal of Climatology, 2018, 38: e921-e938.

[Figure]

**Figure 5. Theil-Sen trend and Mann-Kendall test for ERA5-PWV and GNSS-PWV from 2010 to 2022: (a) Winter Theil-Sen trend, (b) Winter Mann-Kendall test, (c) Spring Theil-Sen trend, (d) Spring Mann-Kendall test, (e) Summer Theil-Sen trend, (f) Summer Mann-Kendall test, (g) Autumn Theil-Sen trend and (h) Autumn Mann-Kendall test.**

Figs. 5(b), (d), (f) and (h) show that PWV change trends in most North America regions are unsignificant in different seasons from 2010 to 2022. Some regions where PWV change trend is significant change constantly in different seasons. Figs. 5 (a) and (b) indicate that the PWV in winter along the Atlantic coast of southeast North America and the northern Gulf Coast has a significant increase trend with the PWV trend of above 3 mm decade$^{-1}$. Figs. 5 (c) and (d) show that the PWV trend in spring is the same as in winter in the northern Gulf Coast. Figs. 5 (e) and (f) indicate that the PWV in summer in the west of Rocky Mountains and Gulf of California has a significant increase trend with the PWV trend of above 3 mm decade$^{-1}$. However, Figs. 5 (c), (d), (e) and (f) indicate that the PWV in spring and summer near the Great Plains and the Great Lakes has a significant decrease trend with the PWV trend of -2 mm decade-1.

Figs. 5 (g) and (h) indicate that the PWV in autumn in the northern Gulf Coast has a significant increase trend with the PWV trend of above 4 mm decade[-1]. The reasons for PWV trend increase are that global warming and increased wind speeds are bringing more water vapor to North America. Water vapor transport is affected by the Appalachian Mountains, so the increase in PWV trend content mainly occurs in southeast of Appalachian Mountains. The reasons for PWV trend increase are that global warming and increased wind speeds are bringing more water vapor to North America. Water vapor transport is affected by the Appalachian Mountains, so the increase in PWV trend content mainly occurs in southeast of Appalachian Mountains. Figs. 5 (a), (b), (g) and (h) indicate that the PWV in autumn and winter in the west of Rocky Mountains has a significant decrease trend with the PWV trend of -1 mm decade[-1].

[Figure]

**Figure 6. (a) Theil-Sen trend and (b) Mann-Kendall trend for ERA5-PWV and GNSS-PWV from 2010 to 2022.**

Meanwhile, Fig. 6 (b) shows that most areas do not significant change, while there is a significant increase along the Atlantic coast region of southeast North America and near the Gulf of California.

[Figure]

**Figure 7. Theil-Sen trend and Mann-Kendall test for ERA5-PWV and GNSS-PWV from 1940 to 2022: (a) Winter Theil-Sen trend, (b) Winter Mann-Kendall test, (c) Spring Theil-Sen trend, (d) Spring Mann-Kendall test, (e) Summer Theil-Sen trend, (f) Summer Mann-Kendall test, (g) Autumn Theil-Sen trend and (h) Autumn Mann-Kendall test.**

Figs. 7 (a), (c), (e) and (d) indicates that PWV in North America shows an increase trend in different seasons from 1940 to 2022. However, the significance of the change trend of PWV in different seasons and different regions is obviously different. The change trend in the east of Rocky Mountains is not significant in any season. Fig. 7 (b) indicates that the PWV in northern Rockies and northern Great Plains shows an extremely significant increase trend, and that the PWV in other most North America shows an unsignificant change trend. Fig. 7 (d) shows that the PWV in most North America except in the southern Rockies and the Gulf Coast with an unsignificant change trend shows an extremely significant increase trend. Fig. 7 (f) demonstrates that the PWV in most North America except near the Rocky Mountains with an unsignificant change trend shows an extremely significant increase trend. In particular, the trend of PWV change in the Gulf of California is 1.2 mm decade$^{-1}$. Fig. 7 (h) depicts that the PWV in most North America except the southern Rocky Mountains and Pacific coast of western North America with an unsignificant change trend shows an extremely significant increase trend. In particular, the trend of PWV change in Gulf of California and Gulf of Mexico is 0.6 mm decade$^{-1}$.

[Figure]

**Figure 8. (a) Theil-Sen trend and (b) Mann-Kendall trend for ERA5-PWV and GNSS-PWV from 1940 to 2022.**

The Theil-Sen trend estimates from 1940 to 2022 in North America are shown in Fig. 8. Fig. 8 (a) indicates that the whole interdecadal trend of PWV is rising in North America from 1940 to 2022. Meanwhile, in Fig. 8 (b), you can see that the trend of PWV remains unchanged east of the Rocky Mountains. The other regions are all experiencing significant increases, and most of them are experiencing extremely significant increases, with the fastest rising interdecadal trend of around 0.60 mm decade$^{-1}$ near the Gulf of California. The main reasons of PWV trend increase include two aspects. On the one hand, as global atmospheric and sea surface temperatures rise, sea surface evaporation increases. On the other hand, the global wind speed is increasing, and large amounts of water vapor are transported to the North American continent. The Rocky Mountains on the west side and Appalachian Mountains of the North American continent block water vapor from the Pacific Ocean, so the increase in water vapor is concentrated in the southwest of the Rocky Mountains, the north of Gulf of Mexico and the southeast of Appalachian Mountains.

The following pages are our point-to-point responses to each of the comments of the anonymous reviewers and Editor in Chief. We look forward to hearing from you at your earliest convenience.

Other specific comments include,

P1L10: why choosing data from 2010 to 2022? Only this period is available for Suominet? The length of 13 yrs is not enough for long-term investigations.

**Reply:** Suominet has provided PWV data since 2006, but data of many stations are missing and incomplete from 2006 to 2009. Therefore, we choose some stations of available and relatively complete data from 2010 to 2022. The GNSS PWV of 13 years mainly are used to verify the long-term change trend agreements of GNSS and ERA5 PWV. ERA5 PWV are used for more longer-term investigations.

P4L98: how did you get PWV from ERA5? More details are needed.

**Reply:** Thank for your suggestions. We have added more detailed discussions about how I get PWV from ERA5, which is shown in the following.

The monthly mean single level PWV and Temperature data from 1940 to 2022 covering N10°~ N 50° and W130°~ W 50° in NetCDF format can be provided from ECMWF (https://climate.copernicus.eu/).

Figure 2: the distribution of GNSS stations is different in different subplots. Why?

**Reply:** I apologize for any confusion caused. The distributions of GNSS stations in all subplots in Figure 2 and other figures have been checked and corrected.

P7L146: 'a good agreement between …', a quantitative analysis is needed rather than just qualitative description. The authors also need to check other sections.

**Reply:** Thank for your suggestions. We have added a quantitative analysis about the EAR5 PWV and GNSS PWV, which is shown in the following.

[Figure]

**Figure 4. (a) bias and (b) RMSE values for ERA5-PWV relative to GNSS-PWV around the North America.**

From Figure 4(a), it can be seen that the bias values of ERA5-PWV and GNSS-PWV mainly focus on the range of 0~2 mm, indicating that the ERA5-PWV is overestimated compared with GNSS-PWV overall. Figure 4(b) shows that most RMSE values of ERA5-PWV and GNSS-PWV are smaller than 3 mm and the large RMSE values are mainly distributed in Gulf of Mexico, which are similar to the distribution pattern of the bias. The results indicate that there is a good agreement between the annual mean GNSS-PWV and ERA5-PWV.

Figure 4: the differences of trends in different seasons need explanations and discussions.

**Reply:** Thank for your suggestions. We have added more detailed discussions about the differences of trends in different seasons, which is shown in the following.

Global warming and increased wind speeds are bringing more water vapor to North America. Water vapor transport is affected by the Appalachian Mountains, so the increase in PWV trend content mainly occurs in southeast of Appalachian Mountains.

Figure 9 and the corresponding discussions are just too simple.

**Reply:** Thank for your suggestions. We have added more detailed discussions about PWV and ENSO in the revised manuscript, which is shown in the following.

Through the findings of previous studies, we found that PWV and Temperature a clear annual oscillation and some of them showed a semi-annual oscillation. Therefore, a linear least-squares sine and cosine model is introduced for the PWV and Temperature time series:

$$x(t) = x_0 + v \cdot t + a_0 \cdot \sin(2\pi \frac{t}{365.25}) + a_1 \cdot \cos(2\pi \frac{t}{365.25})$$

$$+ a_2 \cdot \sin(4\pi \frac{t}{365.25}) + a_3 \cdot \cos(4\pi \frac{t}{365.25})$$

where $x(t)$ is the PWV time series, $x_0$ is the mean value of PWV time series, $v$ is the linear trend and $a_0 - a_3$ are coefficients of sine and cosine for annual and semi-annual terms, respectively. The values of $x_0$, $v$ and $a_0 - a_3$ are estimated based on the month mean values of the PWV dataset found using the least squares method.

[Figure]

**Figure 10. PWV, Temperature and power spectrum from 1980 to 2022.**

The North American monthly mean PWV series from 1980 to 2022 and amplitude based on FFT are shown in Fig. 10. Fig. 10 (b) shows that PWV has obvious annual period term with the amplitude of 9.43 mm and half-year period term with the amplitude of 1.94 mm. The pink and red lines in Fig. 10 (a) are the trend term data and the period term data obtained by using a linear fitting model with periodic terms. The residual term is the original data minus the period term and the trend term and the residual term change may be caused by an ENSO event, which is discussed in Fig 11.

[Figure]

**Figure 11. ONI and PWV residual from 1980 to 2022 in the (a) whole, (b) tropics and (c) temperate zone of North America.**

Fig. 11 (a) demonstrates that super or strong class El Niño events occur during 1982-1983, 1987-1988, 1991-1992, 1997-1998, and 2015-2016 and that PWV residual increases significantly to above 1.5mm from six to ten months after El Nino events.

However, PWV do not rise significantly after the extremely strong ENSO event of 1991-1992, probably due to the 1991 volcanic eruption in the Philippines resulting in the decrease of global temperature. The PWV values decrease significantly is reduced by 1.5~3mm after strong La Niña events occur during 1988-1989, 1998-2000, 2007-2008, and 2010-2011. The results indicate that ENSO events have an important influence on PWV change. There is no significant change in PWV after individual ENSO event and La Niña events, which may be due to the superimposed effects of the Atlantic Oscillation and the Arctic Oscillation. Figs. 11 (b) and (c) show that ENSO-induced water vapor changes range from 3 to 8 mm and from 1 to 2 mm in the tropics (10° N ~ 23.5° N) and temperate zones (23.5° N ~ 50° N), respectively. The results show that the magnitude of ENSO influence on PWV in tropics is higher than that in temperate zone. Figs. 11 (b) and (c) show the response time of PWV to ENSO is shorter in the tropics than temperate zones. The findings indicate that the effect of ENSO on PWV is greater in the tropics than in temperate zones.

---

## Author Comment (AC2)

**Reviewer #2: https://doi.org/10.5194/egusphere-2023-2508-EC1**

I am providing this comment on behalf of a referee, who sends her/his short report only via email and missed to submit it as a comment in the ACP open discussion.

The reviewer's assessment of the paper was that it does not make any significant new scientific contribution, but rather repeats earlier work. Therefore, the referee recommended rejection of the paper. This assessment was confirmed upon examination of previous articles (examples listed below).

References:

[1] Chen, B. and Liu, Z.: Global water vapor variability and trend from the latest 36 year (1979 to 2014) data of ECMWF and NCEP reanalyses, radiosonde, GPS, and microwave satellite, J. Geophys. Res. Atmos., 121, 11,442-411,462, 10.1002/2016JD024917, 2016.

[2] Peng, W., Tongchuan, X., Jiageng, D., Jingmin, S., Yanling, W., Qingli, S., Xin, D., Hongliang, Y., Dejun, S., and Jinrong, Z.: Trends and Variability in Precipitable Water Vapor throughout North China from 1979 to 2015, Adv. Meteorol., 2017, 1-10, 10.1155/2017/7804823, 2017.

[3] Wang, S., Xu, T., Nie, W., Jiang, C., Yang, Y., Fang, Z., Li, M., and Zhang, Z.: Evaluation of Precipitable Water Vapor from Five Reanalysis Products with Ground-Based GNSS Observations, Remote Sens., 12, 10.3390/rs12111817, 2020.

[4] Xu, Y., Ma, L., Zhang, F., Chen, X., and Yang, Z.: Accuracy Analysis of Real-Time Precise Point Positioning—Estimated Precipitable Water Vapor under Different Meteorological Conditions: A Case Study in Hong Kong, Atmosphere, 14, 650, 10.3390/atmos14040650, 2023.

Reply: First of all, thank you very much for giving some references and helpful comments. However, our manuscript is very different from four references. References [3] and [4] mainly focus on evaluation of real time water vapor and PWV from five reanalysis products, rather than PWV trend analysis. References [2] and [4] mainly study the variation trend of water vapor in northern China and across the world, and the time span does not exceed 40 years. The variation trend of water vapor and reasons are different in different countries and regions, and the variation trend of water vapor in north America is absolutely different from China and the globe.

Compared with the above four references, the main innovations of the manuscript are as follows:

(1) We verify and analyze the consistency of long-term water vapor trends between GNSS and ERA5 PWV from 2010 to 2022, which are different from simple accuracy evaluation of several reanalysis.

(2) We analyze water vapor variation trends and variation reasons throughout the year and in different seasons over an 83-year time span in North America, which is more meaningful for long-term climate change studies.

(3) We analyze and discuss the effect of El Nino events and La Niña events on water vapor change.

Reference:

[1] Chen, B. and Liu, Z.: Global water vapor variability and trend from the latest 36 year (1979 to 2014) data of ECMWF and NCEP reanalyses, radiosonde, GPS, and microwave satellite, J. Geophys. Res. Atmos., 121, 11,442-411,462, 10.1002/2016JD024917, 2016.

[2]  Peng, W., Tongchuan, X., Jiageng, D., Jingmin, S., Yanling, W., Qingli, S., Xin, D., Hongliang, Y., Dejun, S., and Jinrong, Z.: Trends and Variability in Precipitable Water Vapor throughout North China from 1979 to 2015, Adv. Meteorol., 2017, 1-10, 10.1155/2017/7804823, 2017.

[3]  Wang, S., Xu, T., Nie, W., Jiang, C., Yang, Y., Fang, Z., Li, M., and Zhang, Z.: Evaluation of Precipitable Water Vapor from Five Reanalysis Products with Ground-Based GNSS Observations, Remote Sens., 12, 10.3390/rs12111817, 2020.

[4]  Xu, Y., Ma, L., Zhang, F., Chen, X., and Yang, Z.: Accuracy Analysis of Real-Time Precise Point Positioning—Estimated Precipitable Water Vapor under Different Meteorological Conditions: A Case Study in Hong Kong, Atmosphere, 14, 650, 10.3390/atmos14040650, 2023.